# Carnitine in Human Muscle Bioenergetics: Can Carnitine Supplementation Improve Physical Exercise?

**DOI:** 10.3390/molecules25010182

**Published:** 2020-01-01

**Authors:** Antonio Gnoni, Serena Longo, Gabriele V. Gnoni, Anna M. Giudetti

**Affiliations:** 1Department of Basic Medical Sciences, Neuroscience and Sense Organs, University of Bari “Aldo Moro”, 70124 Bari, Italy; antonio.gnoni@uniba.it; 2Department of Biological and Environmental Sciences and Technologies, University of Salento, 73100 Lecce, Italy; serena.longo@unisalento.it (S.L.); gabriele.gnoni@unisalento.it (G.V.G.)

**Keywords:** l-carnitine, fatty acid oxidation, muscle energetics, physical exercise

## Abstract

l-Carnitine is an amino acid derivative widely known for its involvement in the transport of long-chain fatty acids into the mitochondrial matrix, where fatty acid oxidation occurs. Moreover, l-Carnitine protects the cell from acyl-CoA accretion through the generation of acylcarnitines. Circulating carnitine is mainly supplied by animal-based food products and to a lesser extent by endogenous biosynthesis in the liver and kidney. Human muscle contains high amounts of carnitine but it depends on the uptake of this compound from the bloodstream, due to muscle inability to synthesize carnitine. Mitochondrial fatty acid oxidation represents an important energy source for muscle metabolism particularly during physical exercise. However, especially during high-intensity exercise, this process seems to be limited by the mitochondrial availability of free l-carnitine. Hence, fatty acid oxidation rapidly declines, increasing exercise intensity from moderate to high. Considering the important role of fatty acids in muscle bioenergetics, and the limiting effect of free carnitine in fatty acid oxidation during endurance exercise, l-carnitine supplementation has been hypothesized to improve exercise performance. So far, the question of the role of l-carnitine supplementation on muscle performance has not definitively been clarified. Differences in exercise intensity, training or conditioning of the subjects, amount of l-carnitine administered, route and timing of administration relative to the exercise led to different experimental results. In this review, we will describe the role of l-carnitine in muscle energetics and the main causes that led to conflicting data on the use of l-carnitine as a supplement.

## 1. Introduction

Carnitine (3-hydroxy-4-*N*-trimethylaminobutyrate) represents an amino acid derivative and a micronutrient that plays a key role in intermediary metabolism with the main function being the transport of long-chain fatty acids from the cytosol to the mitochondrial matrix where fatty acid β-oxidation occurs. Other established functions of carnitine are the preservation of membrane integrity [1], the stabilization of a physiologic coenzyme A (CoASH)/acetyl-CoA ratio in mitochondria, and the reduction of lactate production [2,3].

Carnitine is present in most, if not all, animal species and in several micro-organisms and plants. In the human body, carnitine is mainly found in a free form (free carnitine) and in the form of acylcarnitine esters, a pool of carnitine bounded to various acyl groups that are delivered throughout the body for a wide range of functions [4]. At rest, the skeletal muscle carnitine pool is distributed as 80–90% free carnitine, 10–20% short-chain acylcarnitines (with a number of carbon atoms <10), and <5% long-chain acylcarnitines (with a number of carbon atoms >10) [5].

It has been estimated that the total carnitine content in the human body is about 300 mg/kg, with about 95% stored intracellularly in the heart and skeletal muscle, and the remaining part in the liver, kidney, and plasma [6]. The amount of circulating plasma carnitine accounts for only 0.5% of total body carnitine [7].

Carnitine does not undergo metabolic changes and, therefore, is eliminated as free carnitine in urine. However, a part of carnitine that is not absorbed at the level of the small intestine is completely degraded by bacteria in the large intestine to produce trimethylamine, a quaternary amine that, after enterocyte absorption, is oxidized in the liver by flavin-containing monooxygenase 3 to form trimethylamine-*N*-oxide (TMAO) [8].

In light of the fundamental role of carnitine in fatty acids β-oxidation for energy production, studies have been conducted to understand whether carnitine supplementation can affect skeletal muscle function and athletic performance in healthy individuals [2,9]. There is controversy as to whether or not carnitine administration can improve physical performance. The difference in exercise intensity, the training or conditioning of the subjects, the amount of carnitine administered, the route of administration and the timing of administration relative to the exercise led to different experimental results [10]. In this review, we will summarize the main roles of carnitine in the skeletal muscle energetics and the principal pharmacokinetic characteristics of carnitine in order to highlight the main critical points for carnitine supplementation during exercise.

## 2. Endogenous Synthesis and Cell Transport of l-Carnitine

Humans obtain carnitine mainly from the diet, predominantly from animal-based food products such as red meat, chicken, fish and dairy. Only 25% of carnitine comes from endogenous synthesis [2,11]. Carnitine biosynthesis requires two essential amino acids: l-lysine, which provides the carbon backbone and l-methionine that furnishes the *N*-methyl group [12]. The pathway of carnitine synthesis requires vitamin C, vitamin B6, niacin, and reduced iron as cofactors [13,14]. Lysine residues in some proteins undergo *N*-methylation using S-adenosylmethionine as a methyl donor, forming 6-*N*-trimethyl-lysine residues which are converted in carnitine in four enzymatic steps, namely hydroxylation at carbon 3, aldol cleavage, oxidation of the aldehyde to 4-butyrobetaine and hydroxylation of 4-butyrobetaine at carbon 3 [15]. The last enzymatic step is catalyzed by the enzyme 4-butyrobetaine dioxygenase to yield carnitine. It is generally recognized that the first three reactions of carnitine synthesis are widely distributed in the body but the final reaction is only present in the liver, kidney, and brain [7]. Thus, other tissues depend on carnitine uptake from the circulation.

Carnitine homeostasis is guaranteed by intestinal absorption from the diet, modest endogenous synthesis and efficient renal reabsorption. Intestinal absorption of carnitine occurs by both passive (small intestine and colon) and active (duodenum and ileum) mechanisms [16]. The renal process ensures tubular reabsorption of about 98–99% of filtered carnitine thus conserving a normal carnitine level also in strict vegetarians (vegans) and lacto-ovo-vegetarians [17]. The renal threshold for carnitine excretion is about 50 µmol/L that is about the same value of the normal carnitine plasma concentration [18]. Thus, when plasma carnitine concentration raises, the renal excretion increases whereas the reabsorption decreases, thus maintaining baseline carnitine plasma level [18]. The opposite occurs when the plasma carnitine level is decreased, as in the case of low dietary intake of carnitine [18]. Neither renal reabsorption nor changes in dietary carnitine intake appear to affect the rate of endogenous carnitine synthesis [19,20].

The plasma membrane transport of carnitine is made by the carnitine/organic cation transporters (OCTN), a family of transporters that consists of three isoforms, i.e., OCTN1 (Scarnitine22A4) and OCTN2 (Scarnitine22A5) in humans and animals [21], and Octn3 (Slc22a21) in mice and humans [22]. Among them, OCTN2 is a high-affinity plasma membrane protein that can transport carnitine in a sodium-dependent manner and whose functional defect causes primary systemic carnitine deficiency [23]. OCTN2 is mainly involved in maintaining carnitine homeostasis that results from intestinal absorption, distribution to tissues, and renal excretion/reabsorption. OCTN2 has a high affinity for carnitine and its derivatives and it is present not only at the level of polarized cells of intestine, kidney, placenta and mammary gland but also in other tissues such as liver, heart, testis, skeletal muscle and brain [24,25]. OCTN1, highly expressed at the renal epithelium level and, to a lesser extent, in other tissues, is also involved in carnitine transport but with a lower affinity with respect to OCTN2 [23,24]. Moreover, other transporters such as ATB^0,+^ [26] and carnitine transporter-2 [27] can also carry carnitine.

Among different peripheral tissues, muscle is probably the main target of carnitine transport. About 90–95% of total carnitine is concentrated in muscle, and the ratio between muscle carnitine concentration and plasma concentration is around 50:1 [28].

Carnitine transport defect, associated with mutations in the OCTN2 transporter, causes profound intracellular and plasma carnitine depletion and very high renal carnitine excretion [29]. The consequence of such a defect is the inability of carnitine to perform its functions, particularly in β-oxidation. Metabolic and clinical aberrations associated with the defect in carnitine transport can be prevented by oral supplementation of pharmacological carnitine dosages (100–400 mg/kg/day). Due to the defective OCTN2, this treatment can increase the plasma carnitine level nearly to normal values but does not bring muscle carnitine concentration to normal values [29].

## 3. Role of Carnitine in Mitochondrial Fatty Acid Transport and β-Oxidation

Fatty acids from different sources (diet, endogenous de novo synthesis and adipose tissue hydrolysis) can cross the plasma membrane and enter the muscle cell by fatty acid transport proteins (FATPs), fatty acid translocase (FAT/CD36), caveolins and plasma membrane fatty acid-binding proteins (FABPpm) [30]. Once in the cell, long-chain fatty acids are activated to fatty acyl-CoAs by a family of acyl-CoA synthetases (ACS) identified at the plasma membrane, mitochondria, and lipid droplets [31]. FATPs have also ACS activity [32]. The activation reaction, that requires ATP and CoASH, traps long-chain fatty acyl-CoA thioesters inside the cell and, maintaining intracellular free long-chain fatty acids to low concentrations, allows further fatty acid uptake.

As long-chain acyl-CoA derivatives cannot directly cross the mitochondrial inner membrane, the entry of acyl-CoAs into the mitochondrial matrix for β-oxidation is guaranteed, under normal conditions, by carnitine palmitoyltransferase-1 (CPT-1), located on the external surface of the outer mitochondrial membrane [33]. CPT-1 catalyzes the transfer of acyl groups from acyl-CoAs to carnitine to produce acylcarnitines and free CoASH (Figure 1). The reaction catalyzed by CPT-1 is tightly regulated to control both fatty acid β-oxidation and ketone body formation [34,35]. There are three different isoforms of CPT-1: CPT-1A, CPT-1B, and CPT-1C [35]. CPT-1 A is expressed in the liver, brain, kidney, lung, spleen, intestine, pancreas, ovary and fibroblasts [36]. CPT-1B is the muscle isoform and is highly expressed in skeletal muscle, heart and testis [36]. CPT1-C is the neuron-specific isoform, but its function in neural metabolism remains controversial [37]. CPT-1 is sensitive to inhibition by malonyl-CoA [38]. Since malonyl-CoA represents the product of acetyl-CoA carboxylase, a key enzyme of the cytosolic fatty acid synthesis pathway, malonyl-CoA can reciprocally regulate fatty acid synthesis and oxidation [38]. Thus, a high rate of fatty acid synthesis results in a low rate of fatty acid oxidation, and vice versa.

Once synthesized, acylcarnitines cross the outer mitochondrial membrane, which is permeable to small molecules [39], and translocate into the matrix by the carnitine/acylcarnitine transporter (CACT), a protein of the inner mitochondrial membrane belonging to the mitochondrial carrier protein family [40,41]. CACT transports acylcarnitines in the matrix in exchange for intramitochondrial free carnitine [41,42,43]. Once in the mitochondrial matrix, fatty acyl units are transferred from carnitine to CoASH by carnitine palmitoyltransferase-2 (CPT-2) to form acyl-CoAs that enter the β-oxidation pathway (Figure 1).

The fatty acid β-oxidation, a mitochondrial process regulated by both nutritional and hormonal factors [44,45], involves the repetitive removal of two carbon units, in the form of acetyl-CoA, from the fatty acyl-chain. The enzymes 2-enoyl-CoA hydratase, 3-hydroxyacyl-CoA dehydrogenase and 3-oxoacyl-CoA thiolase are subsequently involved in the fatty acid β-oxidation process to complete the conversion of the acyl-CoA ester into acetyl-CoAs. The last step releases the two-carbon acetyl-CoA and a ready primed acyl-CoA that takes another turn down the spiral. In total each turn of the β-oxidation spiral produces reduced cofactors in the form of NADH (H^+^) and FADH_2_, and one acetyl-CoA. Further oxidation of acetyl-CoAs via the Krebs cycle produces ATP and additional NADH (H^+^) and FADH_2_ [46]. Electrons from NADH (H^+^) and FADH_2_ pass through the electron transport chain, localized at the inner mitochondrial membrane, to oxygen which is reduced to water. During the passage of electrons throughout the different complexes of the electron transport chain is released energy to generate a transmembrane proton gradient which is used to generate ATP (Figure 1). The complete oxidation of a fatty acid produces numerous ATP molecules.

While the transport of long-chain acyl-CoAs, such as palmitoyl-CoA, oleoyl-CoA and linoleoyl-CoA, requires the presence of both CPT-1 and CPT-2, the transport and oxidation of medium- (C6-C12) and short-chain (C4-C6) fatty acids seems largely independent from the carnitine shuttle [47,48].

The hydroxyl group of carnitine can also form esters with acetate, to generate acetyl-carnitine, or with different carboxylic acids, including fatty acids of all chain lengths, to form a wide array of acylcarnitines [49].

## 4. Muscle l-Carnitine Selects Fuels during Exercise

During muscle contraction, aerobic and anaerobic metabolic pathways contribute to the energy supply according to the duration and intensity of muscle effort [50]. There is an inverse relationship between the duration and the intensity of muscle effort, i.e., very intense muscle contractions can be maintained only for a short duration, while less intense contractions can be sustained or repeated for longer periods. Exercise can be classified as low-to-moderate intensity (<70% maximal oxygen consumption, VO_2_max), or high intensity (>75%, VO_2_max) [51]. At low work rates, muscle aerobic metabolism predominates, lactate does not accumulate, and exercise can be sustained. By contrast, at high work rates, lactate accumulates in muscle and blood, and subjects become rapidly fatigued.

Fats and carbohydrates represent the two major energy sources for physical exercise. Either source can predominate, depending upon the duration and intensity of exercise, degree of prior physical conditioning, and the composition of the diet consumed in the days before the exercise [52]. To guarantee a high delivery of these substrates to the muscle, during the passage from a moderate to maximal exercise intensity a higher blood flow to legs has been demonstrated in steady-state 1-leg kicking performed for several minutes [53].

During an exercise with a VO_2_max below 50%, fatty acid oxidation is favored as a source of energy. In this condition, the utilization of energy from fat increases endurance, spares the use of muscle glycogen and delays the onset of fatigue [54]. At a moderate intensity of work (between 50% and 70% VO_2_max), both glucose and fatty acids may be used, with a gradual increase in glucose over fat consumption with incremental exercise intensity. Increasing the intensity of physical activity (above 75%, VO_2_max), fatty acid utilization begins to decline and, above this intensity exercise, glycogen becomes the major fuel supporting ATP synthesis for muscle activities [55]. Glycogen represents a major source for ATP regeneration during prolonged exercise (>1 h) and high-intensity intermittent exercise [56] so that when glycogen stores are limited, exercise cannot continue [57]. Anyway, due to its limited store, glycogen is rapidly depleted under high-intensity endurance exercise and this phase coincides with fatigue [58,59]. However, it must be considered that an inadequate supply of oxygen to the working muscle has been correlated with rapid glycogen depletion via anaerobic glycolysis [60]. During anaerobic glycolysis, lactic acid is produced which promotes acidosis. A strong connection between pH regulation and muscle work capacity has been shown, suggesting that acidosis strongly contributes to fatigue [61].

## 5. How l-Carnitine Can Regulate Fatty Acid Oxidation during Physical Exercise

Different studies have been conducted to explain the regulatory points influencing the decline of fatty acid oxidation in skeletal muscle during high-intensity endurance exercise [62,63,64]. Considering that carnitine is necessary for the transmembrane fatty acid transport, changes in the muscle free carnitine availability may contribute to the regulation of fatty acid oxidation. Generally, intramitochondrial acetyl-CoA can be generated by both fatty acid β-oxidation and by the activity of the multi-enzymatic complex of the pyruvate dehydrogenase, from the glycolytic pyruvate. Mitochondrial acetyl-CoA can be buffered by conversion in acetyl-carnitines by the ACS enzyme. When acetyl-CoA is generated over its metabolism in the tricarboxylic acid cycle, high amount of muscle free carnitine can be entrapped in the form of acetyl-carnitine thus decreasing the free carnitine pool and compromising the catalytic rate of CPT-1 [65].

Indeed, when the glycolytic flux is increased, as during the initial phase of a high-intensity exercise, the amount of pyruvate and then of mitochondrial acetyl-CoA is increased. In the mitochondria, by the activity of the pyruvate dehydrogenase, the glycolytic pyruvate is converted in acetyl-CoA which represents a negative modulator of the pyruvate dehydrogenase. Thus, to allow glycolysis to proceed, acetyl-CoA is bound to carnitine and converted into acetyl-carnitine by acetyl-carnitine synthase enzyme. Actually, during the passage from a low to a high-intensity exercise, muscle acetyl-CoA and acetyl-carnitine increase in parallel [55]. Thus, a high glycolytic flux, reducing muscle free carnitine availability, limits the CPT-1 reaction and ultimately mitochondrial fatty acid import and oxidation. When the glycolytic flux is reduced, as in the case of prolonged exercise or at lower exercise intensity, the reduced supply of glycolysis-derived acetyl-CoA allows saving a greater amount of free carnitine, which can then be used for the transport and oxidation of fatty acids into mitochondria. Thus, it appears evident that muscle total free carnitine availability influences fuel selection during exercise.

However, in contrast to the idea that supplementation of carnitine can accelerate fatty acid oxidation, data from animal models demonstrated that carnitine supplementation during high-intensity exercise increases glucose oxidation at the expense of fatty acid oxidation [66]. The mechanism at the basis of this phenomenon is that, by decreasing the mitochondrial glycolytic acetyl-CoA by acetyl-carnitine synthesis, the pyruvate dehydrogenase enzyme can be activated, thus facilitating complete glucose utilization with a reduction in lactate accumulation. Therefore, the raised availability of carbohydrate during exercise increases glycolysis and pyruvate flux through the pyruvate dehydrogenase with a corresponding decrease in the rate of fat oxidation. This observation should lead to paying attention to the use of carnitine as a supplement to increase the burning of fatty acids during high-intensity exercise, in particular when carnitine administration is preceded by a load of carbohydrates before exercise.

Moreover, a defect in β-oxidation of long-chain fatty acids, mainly associated with a deficiency in very-long-chain acyl-CoA dehydrogenase, an enzyme of the β-oxidation pathway, can result in accumulation of C14-C18 acyl-CoAs in mitochondria [67]. These molecules are converted into acylcarnitine esters to leave the mitochondria. As a result of the increased production of acylcarnitines, blood free carnitine may decrease. Physical exercise in these patients results in a further decrease in free carnitine concentration in skeletal muscle and the appearance of clinical symptoms [67]. It has been widely discussed whether supplementation of exogenous carnitine is advisable to recover intracellular carnitine concentrations in these patients. What emerges from studies is that increased supply of carnitine can result in a further increase, in these patients, of long-chain acylcarnitines compounds associated with possibly lethal heart rhythm disturbance [68].

## 6. Can Carnitine Supplementation Be Useful in Physical Exercise?

Due to the availability of carnitine over-the-counter, the use of carnitine as a supplement is often disproportionate among endurance athletes. Furthermore, since it has been suggested that carnitine saves muscle glycogen and promotes fat oxidation [69], its integration is recommended to lose weight. Carnitine supplementation has been also reported to spare the use of amino acids as energy sources during exercise making them potentially available for new protein synthesis [70]. This notion justifies the use of carnitine to increase muscle mass during endurance exercise. Indeed, a study conducted on dogs demonstrated that supplemented carnitine experienced less protein degradation as a result of exercise [71].

Despite many years of research on the role of carnitine in muscle metabolism, it is yet not completely established whether carnitine supplementation can improve physical performance in healthy subjects. Data on the effect of carnitine supplementation on exercise performance, maximal aerobic capacity, blood lactate response, or substrate utilization during exercise yielded contradictory results (Table 1). The oral administration of 4 g/day of carnitine for 2 weeks was demonstrated to significantly increase VO_2_max in competitive walkers [72]. The same result was obtained by Dragan and coll. in two different studies, one conducted on 40 and the other on 110 top athletes, both orally supplemented with 3 g carnitine for 3 weeks [73,74]. Dragan and coll. also reported that 1 g of carnitine for 6 weeks or 2 g carnitine supplementation for 10 days induced higher performances in 7 junior athletes [75] and 1 g carnitine intravenously furnished ameliorated physical output and muscle contraction [74]. Two grams of carnitine furnished orally before a high-intensity exercise, in moderately trained males, also demonstrated to increase VO_2_max [76]. An effect on mitochondrial respiratory chain enzyme activities was also measured after the oral supplementation of 2 g carnitine for 4 weeks in 14 endurance athletes [77]. Vice versa, 6 g carnitine furnished for a couple of weeks in 8 healthy males failed to influence the VO_2_max and the respiratory exchange ratio [78].

Exercise performance was slightly improved in 9 untrained subjects orally supplemented with 2 g of carnitine for 2 weeks [79]. Besides, Siliprandi and coll. provided evidence that 2 g/day of carnitine furnished 1 h before exercise enhanced high-intensity exercise by increasing pyruvate dehydrogenase activity and preventing lactate accumulation [80]. Instead, 2 g carnitine supplementation 2 h before the start of marathon [81] or 2 g furnished for 7 days [82] did not affect physical performance and recovery after exercise. Similarly, no significant increase in exercise performance was measured after 4g of carnitine supplementation for 3 months to healthy males [83]. The single supplementation of 3 g or 4 g of carnitine to 26 athletes before exercise led to a decrease in lactate and heart rate responses to the running speeds in supplemented groups compared with placebo, suggesting carnitine taken before physical exercise prolonged exhaustion [84].

Contrasting results were obtained on the effect of carnitine supplementation on substrate utilization during exercise. The supplementation of 2 g or 3 g of carnitine for two weeks did not affect substrate utilization during prolonged moderate-intensity exercise or short-duration exercise [85,86]. No effect on substrate utilization was also measured in 10 exercising subjects after oral supplementation of 2 g carnitine for 4 weeks [87] and after 5 g of oral carnitine supplementation for 5 days in 7 moderately trained males [88]. A similar result was obtained by other different studies [81,89,90,91,92]. A certain effect on lipid utilization by muscle during exercise was reported after the oral supplementation of 2 g of carnitine in 10 trained athletes [93]. Interestingly, a recent work conducted on vegetarians demonstrated that 2 g of carnitine supplemented for 12 weeks did not affect muscle functions and energy metabolism of these subjects [94].

The discrepancies in published researches could be related to the features of carnitine’s pharmacokinetic. First of all, it must be considered that, when orally furnished, carnitine bioavailability is only 5–15% [18,95]. Moreover, as the renal threshold for carnitine secretion is near the physiological plasma carnitine concentration, when the plasma concentration of carnitine dominates this threshold, carnitine is promptly eliminated in the urine [18,95]. In fact, after an acute administration of a large amount of carnitine, most of the carnitine is recovered in the urine [91]. Considering the amount of total carnitine content in the body (about 20 g) [9], the low carnitine bioavailability and the large amount of carnitine lost with the urine after integration, a very high dose of carnitine and for long periods should be integrated to obtain a real increase in the amount of muscular carnitine in healthy subjects.

After supplementation, carnitine must be transported from the plasma into tissues. To this respect, it has been reported that muscle, with respect to other tissues, has a much lower net turnover of carnitine [96]; this feature makes muscle, unlike other tissues, particularly refractory to carnitine supplementation. Furthermore, we must consider that, in physiological conditions, carnitine is transported in the muscle against a concentration gradient and that OCTN2 transporter, having a Km value (3–5 µM) below plasma carnitine concentrations (30–50 µM), is saturated at the physiological carnitine concentrations [11,70]. Thus, it is unlikely that an increased plasma concentration of carnitine can cause higher transport of carnitine in the muscle.

Based on these considerations, it is conceivable to predict that oral carnitine supplementation would have little if any effect on muscle carnitine content in humans, and thus on muscle metabolism. Indeed, studies have demonstrated that even if long-term carnitine administration in humans increases plasma carnitine concentrations it does not increase muscle carnitine content [18].

Studies have reported that carnitine can have a protective effect against muscle disruption after strenuous exercise, with a significant reduction in the release of cytosolic proteins such as myoglobin and creatine kinase [70]. Moreover, carnitine supplementation, by increasing the muscle androgen receptor, may improve protein signaling, needed for recovery after exercise [70]. In addition, mitigating oxidative stress during exercise, carnitine can facilitate muscle recovery [70,71].

Other studies reported, both in animals and humans, an effect of carnitine supplementation on vascular function through endothelial function modulation. Indeed, muscle contractile force in dogs was significantly increased and accompanied by an elevated blood flow after infusion with carnitine and in the absence of increased muscle carnitine content and the result was due to an effect on the vasculature surrounding of the muscle [97,98]. Thus, these studies pointed out an effect that is independent of muscle carnitine accretion and energy production. A crossover study demonstrated that 3 weeks of carnitine supplementation increased, with respect to placebo in which decreased, the post-prandial brachial artery flow-mediated dilation [99].

It has to be considered that an increase in muscle carnitine content was obtained supplementing carnitine in the presence of high circulating insulin level (>50 mU/L) and the effect was correlated with an increase in Na^+^/K^+^ pump activity [100,101]. The same result has been also demonstrated with alternative oral nutrients that stimulate insulin secretion such as whey proteins [102].

Finally, it must be considered that carnitine metabolism by gut microbiota produces trimethylamine which is then converted into TMAO in the liver [8]. Indeed, dietary carnitine supplementation can significantly increase serum TMAO levels in both humans and rodents [103]. Painfully, TMAO has been demonstrated to promote atherosclerosis and to increase cardiovascular risk in animals [102]. In human studies, a significant positive correlation has been found between fasting plasma TMAO levels and major cardiovascular events [8,104,105,106,107,108]. Thus, due to the production of TMAO, the beneficial effect of carnitine is controversial and deserves more attention.

Moreover, oral supplementation of carnitine in animals with defects in the very-long-chain acyl-CoA dehydrogenase enzyme can induce significant accumulation in skeletal muscle of acylcarnitine compounds associated with possibly lethal heart rhythm disturbance [68].

## 7. Conclusions

Carnitine is a compound with well-established functions in cellular metabolism, in particular for energetics purposes, as it supports fatty acid transport into mitochondria for β-oxidation and consequent ATP production. It appears that skeletal muscle carnitine availability influences fuel selection during exercise. Indeed, during high-intensity exercise, carnitine availability limits CPT-1 reaction thus reducing fatty acid oxidation. Theoretically, carnitine supplementation should increase carnitine muscle content thus improving fatty acid oxidation and exercise function in healthy humans. However, so far, no scientific basis supports improvement in exercise performance for healthy individuals or athletes after carnitine supplementation. Moreover, considering that carnitine metabolism produces TMAO which has been recently recognized as a novel risk factor for cardiovascular diseases [104,105,106,107,108], the use of uncontrolled amounts of carnitine as supplements must be carefully reviewed.

Since carnitine is often used by athletes with no clear understanding of its effects and risks, it becomes critical to provide information about the characteristics of this compound and its probable harmful effects on health. With the adoption of educational approaches, it will be possible to reduce the risk associated with dietary and nutritional carnitine supplementation, in particular among athletes.

## Figures and Tables

**Figure 1 molecules-25-00182-f001:**
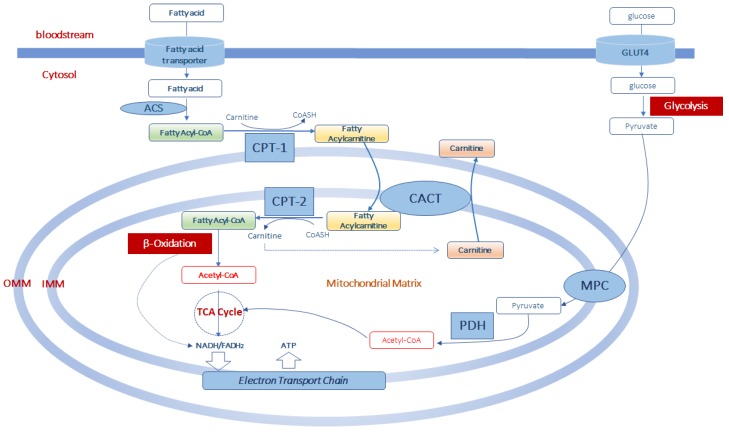
Acetyl-CoA production from fatty acid and glucose metabolism in muscle mitochondria. Fatty acids enter the cells by fatty acid transporters. Once in the cell they are activated to fatty acyl-CoA by acyl-CoA synthetase (ACS) before entering the mitochondria. At the level of the outer part of the outer mitochondrial membrane (OMM), fatty acyl-CoAs are bind to carnitine, by carnitine palmitoyltransferase-1 (CPT-1) activity, to form fatty acylcarnitine derivatives which diffuse through the outer mitochondrial membrane. Thus, the formed fatty acylcarnitines are transported across the inner mitochondrial membrane (IMM) via carnitine-acylcarnitine translocase (CACT). In the mitochondrial matrix, CPT-2 converts fatty acylcarnitines back to fatty acyl-CoAs, which enters the β-oxidation pathway, and to free carnitine which can exit from the mitochondria in exchange with other acylcarnitines through CACT. Mitochondrial acetyl-CoA is generated from both β-oxidation of fatty acids and from pyruvate. Pyruvate is formed in the glycolytic pathway form glucose which enters the muscle cell via the glucose transporter type 4 (GLUT4). Pyruvate is transported in the mitochondrial matrix by the pyruvate carrier (MPC) of the inner mitochondrial membrane. Once in the mitochondria, pyruvate is converted into acetyl-CoA throughout the complex of the pyruvate dehydrogenase (PDH). This acetyl-CoA, together with that formed in the β-oxidation pathway can enter the tricarboxylic acid cycle (TCA cycle) to produce equivalent donors in the form of NADH (H^+^) and FADH_2_ which are oxidized in the mitochondrial electron transport chain to produce ATP.

**Table 1 molecules-25-00182-t001:** Effect of carnitine supplementation on muscle energetics and exercise performance.

Study	Population	Daily Carnitine Dose and Treatment Duration	l-Carnitine Effects
Marconi et al., 1985 [72]	6 competitive walkers	4g orally, 2 wks	Slight but significant increase in VO_2_max.
Greig et al., 1987 [79]	9 untrained subjects	2 g orally, 2 wks	Very little benefit to exercise performance.
Dragan et al., 1987 [73]	40 top athletes	3 g orally, 3 wks	Increase in VO_2_max.
Dragan et al., 1988 [75]	7 junior athletes	1 g orally, 6 wks and 2 g, 10 d	Supplemented group obtained higher performances.
Oyono-Enguelle et al., 1988 [87]	10 exercising subjects	2 g orally, 4 wks	No distinct increase of the relative participation of endogenous lipids in the fuel supply.
Soop et al., 1988 [88]	7 moderately trained males	5 g orally, 5 d	Carnitine does not influence muscle substrate utilization either at rest or during prolonged exercise.
Dragan et al., 1989 [74]	110 top athletes	1 g intravenously (single dose) 3 g orally, 3 wks	Single dose: beneficial effects on physical output, lipid metabolism and muscular function (contraction). 3 weeks treatment: beneficial effects on the lipid metabolism and VO_2_max.
Gorostiaga et al., 1989 [93]	10 trained athletes	2 g orally, 4wks	Increased lipid utilization by muscle during exercise.
Siliprandi et al., 1990 [80]	10 moderately trained males	2 g orally 1 dose 1 h before exam	Stimulation of PDH activity, and decrease in plasma lactate and pyruvate.
Vecchiet et al., 1990 [76]	10 moderately trained males	2 g orally, before high-intensity exercise	Increased VO_2_max.
Wyss et al., 1990 [89]	7 healthy males	3 g orally, 7 d	Lower rate of carbohydrate transformation during hypoxia.
Huertas et al., 1992 [77]	14 athletes	2 g orally, 4 wks	Increase in respiratory-chain enzyme activities in the muscle.
Decombaz et al., 1993 [90]	9 healthy males	3 g orally, 7 d	No influence of l-carnitine on muscle metabolism.
Trappe et al., 1994 [82]	20 male swimmers	2 g orally, 7 d	No differences in performance times were observed between trials or between groups.
Brass et al., 1994 [91]	14 healthy males	185 µmol/kg intravenously	l-carnitine administration has no significant effect on fuel metabolism during exercise in humans.
Vukovich et al., 1994 [78]	8 healthy males	6 g orally, 7–14 d	No differences in VO_2_max and respiratory exchange ratio.
Colombani et al., 1996 [81]	7 male subjects	2 g orally, 2 h before the start of marathon and after 20 km run	l-carnitine does not affect the metabolism and the physical performance of the endurance-trained athletes during the run and did not alter their recovery.
Wachter, et al., 2002 [83]	8 healthy males	4 g orally, 3 months	l-carnitine supplementation is not associated with a significant increase in physical performance.
Broad et al., 2005 [92]	15 trained males	3 g orally, 4 wks	No effect on substrate utilization or endurance performance.
Broad et al., 2008 [85]	20 active male athletes	2 g orally, 2 wks	No effect on fat, carbohydrate, or protein contribution to metabolism during prolonged moderate-intensity cycling exercise.
Broad et al., 2011 [86]	15 athletes	3 g orally, 15 d	l-carnitine induces changes in substrate utilization in metabolically active tissues but it does not affect whole-body substrate utilization during short-duration exercise.
Orer et al., 2014 [84]	26 athletes	12 received 3 g orally 14 received 4 g orally	Both 3 g and 4 g of l-carnitine taken before physical exercise prolonged exhaustion.
Novakova et al., 2016 [94]	16 vegetarians and 8 omnivores	2 g orally, 12 wks	l-carnitine supplementation does not affect muscle function and energy metabolism in vegetarian.

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
