# Peer review of "Carnitine in Human Muscle Bioenergetics: Can Carnitine Supplementation Improve Physical Exercise?"

_molecules, 2020, doi:10.3390/molecules25010182_

Round 1

Reviewer 1 Report

This is an extremely thorough and well written review on carnitine metabolism and it's role in improving muscle physiology and exercise tolerance. It presents a balanced and appropriately referenced view of a subject that is quite controversial in the field. 

Specific comments.

I would suggest the use of "carnitine" rather than LC throughout the manuscript. The abbreviation is distracting and isn't really necessary.

Abstract, line 15 should say "contains high amounts of carnitine"

Page 3, Section 3: Please differentiate between long and medium chain fatty acid oxidation as the latter does not rely on carnitine for import into mitochondria.

Page 5, section 5: consider adding information that pretreatment of patients with long chain fatty acid oxidation disorders decreases the accumulation of long chain acylcarnitine intermediates, indicating a suppression of long chain fatty acid oxidation with decreased depletion of carntine

Page 6, line 229: Delete "madcap" 

Page 7, line 244: fix reference "Dragan and Coll" (delete period)

Page 7, line 253: suggest "exercise was slightly improved" rather than ameliorated

Page 8, lines 288-291: This statement is probably true in normal physiology but not carnitine deficiency states, where supplementation is therapeutic. Please clarify. 

Author Response

I would suggest the use of "carnitine" rather than LC throughout the manuscript. The abbreviation is distracting and isn't really necessary. According with the reviewer suggestion we substituted, throughout the manuscript, LC with the word “carnitine” Abstract, line 15 should say "contains high amounts of carnitine" Done Page 3, Section 3: Please differentiate between long and medium chain fatty acid oxidation as the latter does not rely on carnitine for import into mitochondria. According with the reviewer suggestion we specified that long-chain fatty acids enter mitochondria by CPT-1 activity (line 126). Page 5, section 5: consider adding information that pretreatment of patients with long chain fatty acid oxidation disorders decreases the accumulation of long chain acylcarnitine intermediates, indicating a suppression of long chain fatty acid oxidation with decreased depletion of carnitine In the revised version of the manuscript, we treated the defect in fatty acid β-oxidation and the role that carnitine supplementation can have in these patients, particularly during exercise (page 7 lines 261-270). Moreover, additional voices have been added in the “bibliography” section (references 67 and 68). Page 6, line 229: Delete "madcap"  Done Page 7, line 244: fix reference "Dragan and Coll" (delete period) Done Page 7, line 253: suggest "exercise was slightly improved" rather than ameliorated Done Page 8, lines 288-291: This statement is probably true in normal physiology but not carnitine deficiency states, where supplementation is therapeutic. Please clarify.  According with the reviewer suggestion we introduced information about the effect on supplementation of carnitine in the primary defects of carnitine transport (lines 105-111). Moreover an additional bibliographic voice was also added ([29]).

Reviewer 2 Report

The authors give a comprehensive review about carnitine supplementation and muscle function.

I have the following comments and suggestions:

Line 52: change adsorption to absorption Line 95: change lesser to a lesser extent Legend to Fig. 1: In line 112, the authors claim that CPT1 is expressed in the outer part of the outer mitochondrial membrane. Here, they write that longchainacyl-CoAs have to diffuse across the outer mitochondrial membrane, where they are converted to the corresponding acylcarnitine by CPT1. This suggests that CPT1 is located in the inner part of the outer mitochondrial membrane. The authors should be consistent on this point. Line 173: Classification of work intensity: Here, the authors say that low work intensity is <70% VO2max and high work intensity >75% VO2max. In the section starting at line 183, the definition is different (<50% VO2max low, 50-75% VO2max moderate and >75% VO2max high). This should be clarified. Line 195: In this condition … This sentence should be corrected. Section starting at line 200: Why should the carnitine content in the mitochondrial matrix affect CPT1 activity, when CPT1 is located in the outer mitochondrial membrane? Line 204: Even if the mitochondrial pool is in the range of 10% of the cellular pool in skeletal muscle, the concentration in the mitochondria may still be high enough for saturation of the enzymes/transporters using carnitine as a substrate (consider also the mitochondrial volume fraction). Furthermore, regarding CPT1, the cytosolic carnitine content is probably more important due to its localization in the outer mitochondrial membrane. Line 215: As discussed above, the cytosolic carnitine concentration may be more important for CPT1 activity than the mitochondrial concentration. Line 286: I consider it as unlikely (not likely as stated) that carnitine supplementation increases the skeletal muscle carnitine content. You may want to mention also the Km of OCTN2 for carnitine, which is in the range of 10 µM. OCTN2 is saturated at physiological plasma carnitine concentrations. Line 309: I would say that carnitine supplementation should theoretically not increase the skeletal muscle carnitine concentration, due to the large concentration gradient which makes diffusion impossible and due to saturation of OCTN2 at physiological carnitine concentrations. This is different at low plasma carnitine concentrations (e.g. patients with primary carnitine deficiency) and may be different for acylcarnitines with a low concentration in skeletal muscle. There studies about recovery from exercise, where carnitine may have an effect. Carnitine may not have to enter skeletal muscle to be effective in this case. You may want to add a section about that.

Author Response

Line 52: change adsorption to absorption Done Line 95: change lesser to a lesser extent Done Legend to Fig. 1: In line 112, the authors claim that CPT1 is expressed in the outer part of the outer mitochondrial membrane. Here, they write that longchainacyl-CoAs have to diffuse across the outer mitochondrial membrane, where they are converted to the corresponding acylcarnitine by CPT1. This suggests that CPT1 is located in the inner part of the outer mitochondrial membrane. The authors should be consistent on this point. We thank the reviewer for the note. According with the suggestion we corrected the legend to Fig.1 Line 173: Classification of work intensity: Here, the authors say that low work intensity is <70% VO2max and high work intensity >75% VO2max. In the section starting at line 183, the definition is different (<50% VO2max low, 50-75% VO2max moderate and >75% VO2max high). This should be clarified. Thanks to the reviewer suggestion we better defined the range of work intensity in the manuscript. Line 195: In this condition … This sentence should be corrected. Done Section starting at line 200: Why should the carnitine content in the mitochondrial matrix affect CPT1 activity, when CPT1 is located in the outer mitochondrial membrane? Line 204: Even if the mitochondrial pool is in the range of 10% of the cellular pool in skeletal muscle, the concentration in the mitochondria may still be high enough for saturation of the enzymes/transporters using carnitine as a substrate (consider also the mitochondrial volume fraction). Furthermore, regarding CPT1, the cytosolic carnitine content is probably more important due to its localization in the outer mitochondrial membrane. Line 215: As discussed above, the cytosolic carnitine concentration may be more important for CPT1 activity than the mitochondrial concentration. We agree with the reviewer for this appoint. We changed statements in accordance with a major effect, on CPT-1 activity, by muscle carnitine pool more than the mitochondrial one (lines 222-231). Line 286: I consider it as unlikely (not likely as stated) that carnitine supplementation increases the skeletal muscle carnitine content. You may want to mention also the Km of OCTN2 for carnitine, which is in the range of 10 µM. OCTN2 is saturated at physiological plasma carnitine concentrations. We followed the reviver’s indication and added indications about the Km value of OCTN2 transporter and the effect that an increased plasma carnitine can have on the muscle carnitine level (lines 331-334). Line 309: I would say that carnitine supplementation should theoretically not increase the skeletal muscle carnitine concentration, due to the large concentration gradient which makes diffusion impossible and due to saturation of OCTN2 at physiological carnitine concentrations. This is different at low plasma carnitine concentrations (e.g. patients with primary carnitine deficiency) and may be different for acylcarnitines with a low concentration in skeletal muscle. There studies about recovery from exercise, where carnitine may have an effect. Carnitine may not have to enter skeletal muscle to be effective in this case. You may want to add a section about that. We thank the reviewer for this note. Now we introduced some sentences regarding the effect of carnitine supplementation in subjects with defect in carnitine metabolism (lines 105-110) and in muscle recovery after exercise (lines 344-356).
